

# Wintertime Evolution of Landfast Ice Stability in Alaska from InSAR

Andrew H. Einhorn[1,2], Andrew R. Mahoney[1]

[1]Geophysical Institute, University of Alaska Fairbanks, Fairbanks, 99775, United States of America
[2]Insitute for Marine and Antarctic Studies, University of Tasmania, Hobart, 7001, Australia

*Correspondence to*: Andrew Einhorn (andrew.einhorn@utas.edu.au)

## Abstract

Landfast ice in Alaska is experiencing rapid changes in extents and duration, impacting the safety and utility of the ice for Arctic coastal communities. Current datasets of landfast ice only distinguish landfast ice from mobile pack ice, omitting

crucial information regarding the relative safety within landfast ice. InSAR (Interferometric Synthetic Aperture Radar) holds promise for identification of landfast ice and measurement of cm-scale deformation from a spaceborne sensor. We use two properties of interferometry: coherence to identify areas of landfast ice, and the interferometric phase gradient to approximate a new metric called apparent strain ($\epsilon_a$) which acts as a proxy for estimating the relative stability of the landfast ice. Apparent strain is described as the horizontal gradient of interferometric phase in the line-of-sight displacement. We built on a previous

study by Dammann et al. (2019) by assigning quantitative apparent strain values to identify 3 distinct stability classifications of landfast ice: Bottomfast ($\epsilon_a < 1.0 \times 10^{-5}$), Stabilized ($1.0 \times 10^{-5} \leq \epsilon_a \leq 2.3 \times 10^{-5}$), and Not stabilized ($\epsilon_a > 2.3 \times 10^{-5}$). The monthly average apparent strain decreases as the season progresses, achieving the maximum stability in April or May depending on the region. This study introduces a novel approach to identify the relative stability for areas of landfast ice using InSAR. These findings have implications for enhancing the safety and planning of activities on landfast ice for Arctic coastal

communities.

## 1. Introduction

Landfast ice, also commonly referred to as fast ice or shorefast ice, is sea ice that has become fastened to the coast and remains stationary for a period (Barry et al., 1979). Landfast sea ice is the most commonly encountered form of sea ice due to its proximity to Arctic coastal communities and its relative safety compared to drifting sea ice. Members of Arctic coastal

communities use landfast ice for subsistence hunting and intercommunity travel, among other uses (e.g., George et al. 2004; Laidler et al. 2009). Landfast ice also serves as a habitat for marine mammals and shore birds (Laidre et al. 2015; Lovvorn et al. 2018) and can be used for industrial purposes (Bieniek et al. 2022; Masterson 2009). As a rigid barrier between the ocean and land, landfast ice can mitigate coastal erosion (Hošeková et al. 2021) and modify large-scale circulation patterns (Itkin et al. 2015).

The fundamental property of landfast ice that allows it to perform all these roles in the Earth system is its attachment to the land. Mahoney et al. (2006) proposed that this attachment could be determined from remote sensing data using two criteria: contiguity with the coast and immobility over time. Coastal contiguity can be determined from any single image with





sufficient resolution to observe meaningful leads of open water, but determination of immobility requires at least two images of the same area acquired at different times. For example, Mahoney et al. (2007; 2014) analyzed triplets of co-located synthetic

aperture radar (SAR) data spanning approximately 20 days to identify landfast ice in Alaska, while Fraser et al. (2009; 2012; 2021) and Cooley and Ryan (2024) derived landfast ice extent from 20- and 30-day composites of cloud-free moderate resolution imaging spectrometer (MODIS) data. Landfast ice is also identified in operational ice charts based on analysis of multiple images utilizing a range of satellite based sensing methods (World Meteorological Organization 2024).

Most remote sensing-based analyses of landfast ice extent consider only the binary presence or absence of landfast ice,
omitting other characteristics that might provide information about potential hazards associated with activities on the ice. Here, we apply Interferometric Synthetic Aperture Radar (InSAR) techniques to map and classify landfast sea ice along the northern coast Alaska. Building on previous work (Dammann et al. 2019; Meyer et al. 2011; Pratt 2022) we present an automatable methodology to map the extent landfast ice and identify zones of differing stability within it and we use these to understand how the stability of landfast ice evolves over the winter.

## 45 2. Data and Methods

### 2.1. Study Area

Our study area was chosen to coincide with that used for the development of the EM2024 landfast ice climatology (see section 2.2) and includes waters of the U.S. Arctic Outer Continental Shelf and adjacent waters in Canada and Russia. The coastlines extend from just west of Neshkan on the Chukotka Peninsula to the easternmost point of the Russian mainland in
the Bering Strait and from the westernmost point on mainland Alaska, the Iñupiat village of Wales, to the Mackenzie Delta in Canada. To conform with prior analyses of landfast ice (Mahoney et al. 2007; Mahoney et al. 2014, Mahoney et al. 2024), our study area is divided into two regions (Fig. 1). The western region lies entirely within the Chukchi Sea and hereinafter is referred to as the Chukchi region. The eastern region includes a smaller area of the northern Alaska Chukchi coast but is otherwise contained within the Beaufort Sea and hereinafter is referred to as the Beaufort region. The western and eastern
regions about each other at a point just west of Wainwright Alaska.



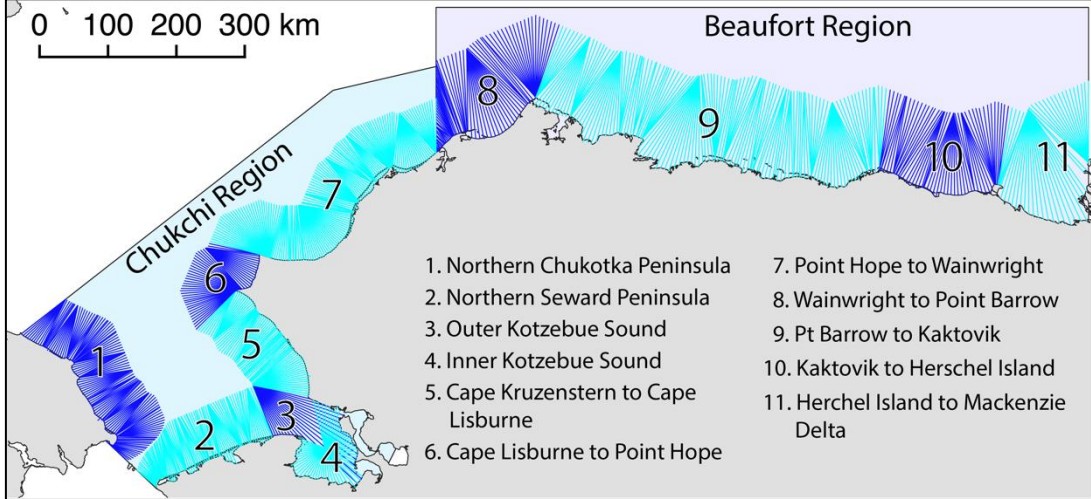

**Figure 1. Spatial extent of study area and coverage of the coast vectors used to measure landfast ice width.**

## 2.2. EM2024 landfast ice climatology

The EM2024 landfast ice climatology (named after the same authors as this study and the year it was released) is a gridded daily dataset of landfast ice extent covering the study area described in section 2.1 and extending from 1996-2023. A detailed description of the creation and results of the EM2024 dataset are reported by Mahoney et al. (2024) but in brief, it extends a previous SAR-based dataset of the same study area for the years 1996-2008 (Mahoney et al. 2014) using data derived from ice charts produced by National Ice Center (NIC) and the National Weather Service's Alaska Sea Ice Program (ASIP). Here, we use landfast ice width data derived from the EM2024 dataset using the SLIEalyzer toolbox ([https://github.com/armahoney/SLIEalyzer](https://github.com/armahoney/SLIEalyzer)), which computes the distance from the coast to the seaward landfast ice edge (SLIE) along a specified set of approximately coast-normal vectors, as illustrated in Figure 1. This approach provides a consistent spatial reference frame with which to measure the seasonal and inter-annual variations in landfast ice almost continuously at 8892 different locations along the coast (6956 in the Chukchi region and 1936 in the Beaufort), spaced approximately every 200 m. It also allows us to calculate average locations of the SLIE over different time periods.

## 2.3. InSAR-based detection of landfast ice

The term InSAR describes a signal processing technique for calculating the phase difference between two radar signals acquired from similar locations in space at separate times. Here, we use pairs of Sentinel-1 interferometric wide (IW) beam images acquired 12 days apart with a spatial baseline of <300 m. These image pairs were selected using the Short Baseline Subset tool within the Alaska Satellite Facility (ASF)'s Vertex portal. 14 IW reference scenes were required to provide complete coverage of both regions in the study area. In total, 2,084 SAR pairs were identified between the months of November and July for the period from March 2017 to July 2022. Multi-looking by 20x4 resulted in pixel size of 80×80 m. We used



ASF's Hybrid Pluggable Processing Pipeline (HyP3) toolbox (Hogenson et al., 2016) to produce interferometric data products in GeoTIFF format. The phase difference between SAR images is often called the interferometric phase, $\phi$, and primarily results from the effects of surface topography when viewed from two different positions, or line-of-sight surface displacement that occurred between image acquisitions. Over sea ice, where topographic expression is generally <10 m, variations in $\phi$ can be expected to be dominated by surface motion for the 12-day short-baseline image pairs used here (e.g., Dammann et al. 2016).

To obtain a useful value of $\phi$, the surface being imaged must remain sufficiently coherent between acquisitions. Interferometric coherence describes the signal similarity, or correlation, between coregistered pixel neighborhoods in the paired images, with a value of 1 indicating perfect correlation and a value of 0 indicating complete decorrelation (Moreira et al. 2013). In the context of sea ice, coherence loss is most commonly caused by ice drift, which is typically on the order of km per day and results in the ice surface comprising a pixel to change completely within a few minutes. There are other processes unrelated to motion that reduce coherence, but if sea ice maintains coherence between image acquisitions it can be assumed to be landfast. Meyer et al. (2011) applied this concept to 45-day repeat Advanced Land Observing Satellite (ALOS) Phased Array L-band Synthetic Aperture Radar (PALSAR) data and determined that pixels with a normalized coherence value greater than 0.1 were landfast ice.

Following the work of Meyer et al. (2011), we apply the same coherence threshold identify landfast ice using from C-band Sentinel-1 imagery, under the assumption that the increased coherence from a shorter repeat orbit interval at least partially offset the reduced coherence effect of a shorter wavelength. We also apply morphological opening and closing filters to the thresholded imagery to remove small "islands" of high coherence amongst drifting ice or open water and "pinholes" of low coherence within the landfast ice. We grouped the filtered and thresholded coherence images by calendar month according to the acquisition date of the primary image in each SAR pair and mosaicked them to create monthly images of InSAR-derived landfast ice extent for each subregion. This allowed use the SLIEalyzer toolbox to obtain measurements of landfast ice width that are directly comparable with monthly means derived from the EM2024 dataset.

## 2.4. Calculation of apparent strain from phase gradient

For short-baseline 12-day SAR pairs over sea ice, we can assume that variations in $\phi$ are dominated by variations in line-of-sight surface motion. $\phi$ is insensitive to motion perpendicular to the line of sight, but Fedders et al. (2024), has shown that it is possible to estimate 2-dimensional horizontal strain of sea ice from the phase gradient provided the phase slope is largely planar and the mode of deformation (e.g., radial divergence or rotation) is known. However, in general, derivation of 2-D or 3-D surface motion requires phase information from multiple look directions. Instead, we define the term apparent strain, $\epsilon_a$, to describe the magnitude of the horizontal gradient in line-of-sight displacement:

$$\epsilon_a = \frac{\lambda}{4\pi} |\nabla\phi| \quad (1)$$

Where $\lambda$ is the SAR wavelength (5.64 cm) and $|\nabla\phi|$ is the magnitude of the phase gradient, given by:




$$|\nabla \phi| = \sqrt{\frac{\partial \phi}{\partial x}^2 + \frac{\partial \phi}{\partial y}^2} \qquad (2)$$

Since $\phi$ is a cyclic quantity that wraps over an interval of $2\pi$, we calculate $\nabla \phi$ following the approach described by Libert et al. (2022) whereby $\phi$ is first converted to a complex quantity, $\phi^*$, with continuous real and imaginary components:

$$\phi^* = e^{i\phi} = cos\phi + i\,sin\phi \qquad (3)$$


$$\frac{\partial \phi}{\partial x} = \angle \left( \frac{\partial \phi^*}{\partial x} \right) \qquad (4)$$

$$\frac{\partial \phi}{\partial y} = \angle \left( \frac{\partial \phi^*}{\partial y} \right) \qquad (5)$$

Where $\angle$ indicates the argument of the complex exponent.

We approximate the partial derivatives in (4) and (5) using finite differences across a 4-pixel window and insert into (2) to derive the magnitude of the phase gradient magnitude. Apparent strain, $\epsilon_a$, is then calculated using (1) and should be

interpreted as the minimum net strain that occurred between SAR image acquisitions, since it only measures deformation along the satellite's line of site. However, since it only represents displacement between two snapshots in time, it may underestimate the maximum strain that occurred during the 12-day period. As with the coherence images, we grouped $\epsilon_a$ results by month according to the date of the primary image in each SAR pair and mosaicked results to create images of monthly average $\epsilon_a$ for each of the Chukchi and Beaufort regions. Pixels with a coherence <0.1 were excluded from the results.

**2.5. Spatial masks for defining apparent strain thresholds for stability classes**

Dammann et al. (2019) used InSAR to define three classes of stability within landfast ice, based on a qualitative analysis of the density and orientation of interferometric fringes: bottomfast; stabilized; and non-stabilized. Bottomfast ice is the most stable of these classes and occurs in water shallow enough where the entire water column freezes, and the sea ice is frozen too or is resting on the seafloor. This depends on the ice thickness (or, more precisely, the draft of the ice), but in our study region

bottomfast ice will be found in waters up to approximately 1.5 m deep (Pratt 2022). In interferograms, bottomfast ice was identified based on "*No identifiable phase difference from the adjacent land*". Stabilized landfast ice is found seaward of the bottomfast ice zone where the ice is floating but held in place by islands or grounded pressure ridges. Dammann et al. (2019) identified stabilized ice based on "*Poorly defined, widely spaced fringes, or abruptly reduced fringe spacing compared to offshore ice*". Lastly, non-stabilized ice includes all landfast seaward of any island or grounded ridge and was identified

according to "*Well-defined fringe orientation or patterns*".

To quantitatively define apparent strain thresholds associated with each of these stability classes, we created the following three non-overlapping spatial masks within the Beaufort study region:



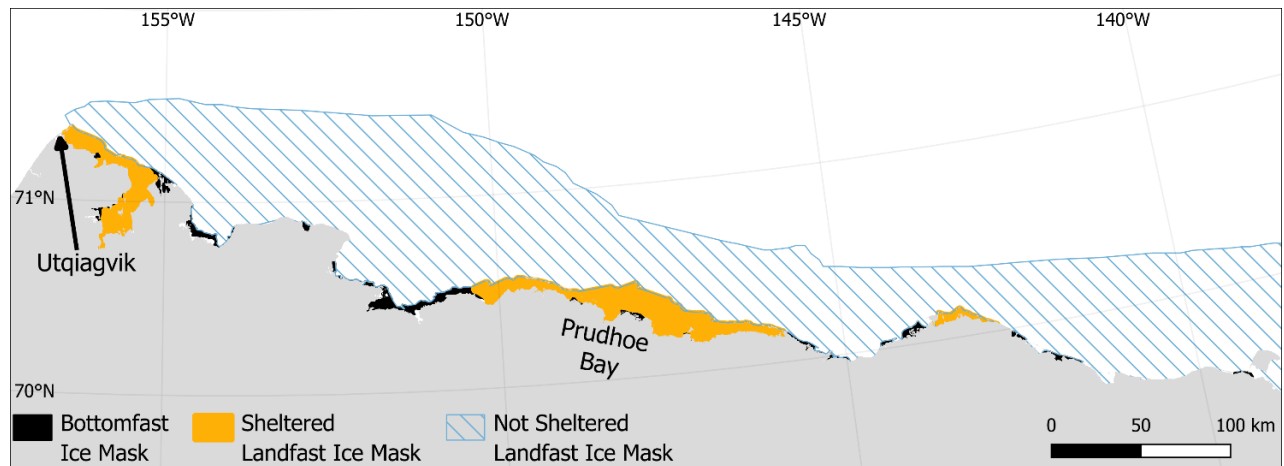

**Figure 2: Spatial extent of bottomfast, sheltered, and not sheltered landfast ice masks**

1.  Bottomfast ice mask. This corresponds to the area identified as bottomfast by Dammann et al. (2019) (black in Fig. 2). The maximum extent of bottomfast ice extent along the Beaufort coast varies little enough between seasons (Pratt 2022) that we can use the bottomfast ice extent defined during April of 2017 by Dammann et al. (2019) to identify the areas which are likely bottomfast ice during April of all seasons.

2.  Sheltered ice mask: This is defined by the area seaward of the bottomfast ice mask and shoreward of any barrier islands (orange areas in Fig. 2). This represents floating landfast ice that can be expected to be sheltered from dynamical interaction with pack ice by the islands and should therefore be representative of stabilized ice.

3.  Not-sheltered ice mask: This is the area seaward of any barrier islands (light blue hashed areas in Fig. 2) and will include all the landfast ice subject to dynamical interaction with pack ice. However, it may also include grounded pressure ridges and so we expect the non-sheltered ice mask to represent both non-stabilized and stabilized landfast ice.

## 3. Results

### 3.1. Landfast ice extent from coherence thresholds

Application of the coherence threshold described in section 2.3 to 2,084 SAR image pairs allowed us to produce 24 monthly images of landfast ice for each month between December and May during the 4 landfast ice seasons between 2017 and 2021. We then used the SLIEalyzer toolbox (section 2.2) to derive the average position of the SLIE over all 4 seasons for each month for comparison with the equivalent results derived from the EM2024 dataset from these. For example, Fig. 3 compares positions of the InSAR-derived mean landfast ice extent during April (cyan) line with that derived from the EM2024





dataset (blue) for the period 2017 –2021. This shows generally good agreement, with no consistent bias across the study region for this month.

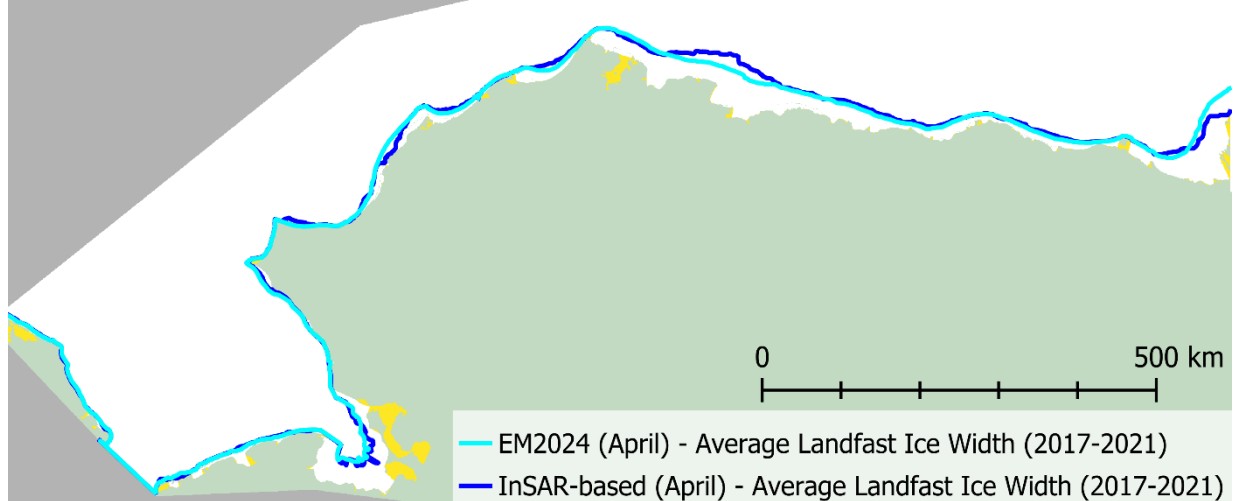

**Figure 3: Mean SLIE position for the month of April by the EM2024 dataset (cyan) and InSAR-based method (blue) from 2017 –2021. Yellow regions indicate "shadow" zones outside the domain of the coast vectors (see Fig. 1)**

To compare the InSAR-derived landfast ice extent against the EM2024 dataset for all months from December to May, we calculated the difference in landfast ice width for each coast vector (Fig. 4). Here, the difference (blue line) is calculated by subtracting the InSAR-derived width from the EM2024 values. The normalized difference (gray line) is derived by dividing the difference by the EM2024 width. During December, mean landfast ice width was zero throughout much of the study region and both datasets generally agree where landfast ice had not yet formed, but small differences in width result in large, normalized differences. However, where landfast was present (primarily in the Beaufort region), the InSAR-derived results tend to underestimate the width. This is illustrated by the tendency of the blue line to lie below the x-axis in Fig. 4a.





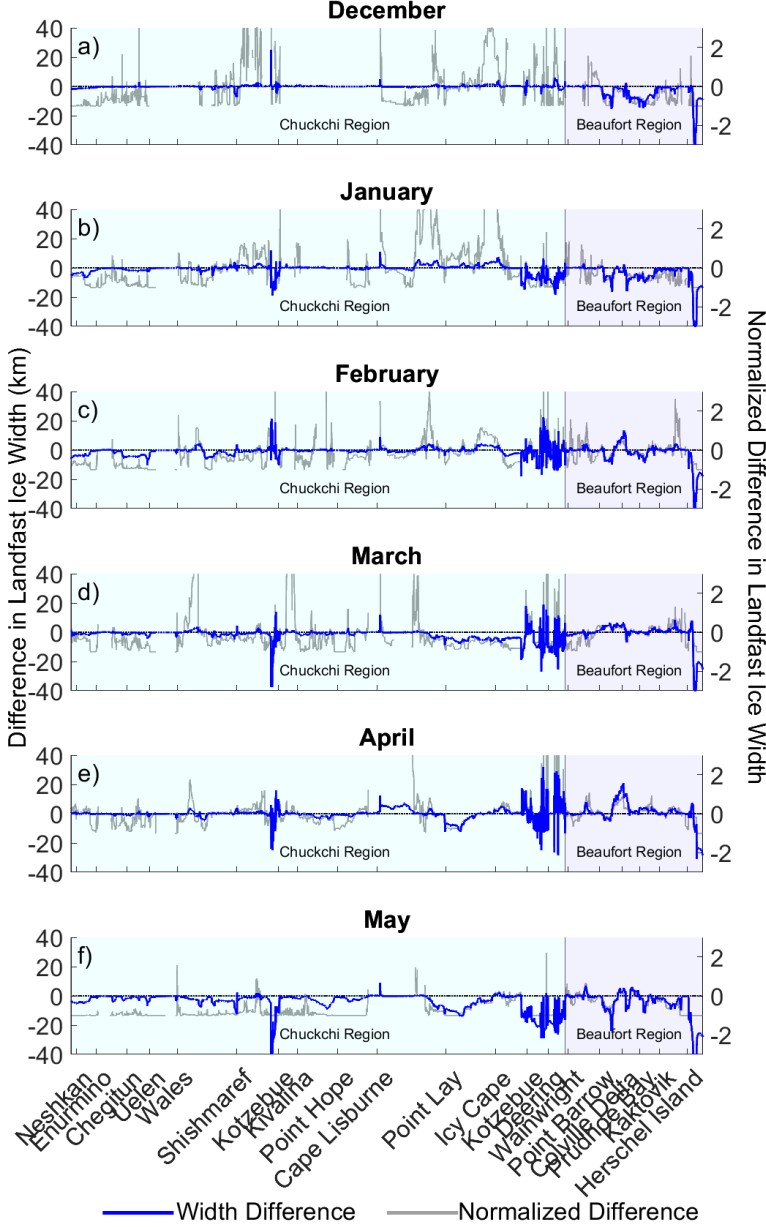

**Figure 4:** Difference in landfast ice width at each coast vector from 2017–2021 between the InSAR-derived results and the EM2024 dataset. Positive values indicate that the InSAR method overestimate landfast ice extent relative to the EM2024 dataset. Cyan shaded regions were within the Chukchi region and blue shaded regions were the Beaufort region.



There is a similar pattern for the month of January (Fig. 4b), with the InSAR-based approach still tending to underestimate the width of landfast ice as it expands throughout both study regions. However as the season progresses through February, March, and April, the relationship becomes more variable along the coast, with the InSAR-based approach indicating over 10 km more landfast ice than the EM2024 dataset at certain coast vectors (Fig. 4b-e). Also, there is notable spatial variability in Kotzebue Sound during these months, with the difference in landfast ice width changing from $<-10$ km to $>10$ km and back again over the span of just a few coast vectors. The cause of this variability is not certain, but examination of the parent SAR imagery and interferograms over Kotzebue Sound indicate that the surface of the ice loses coherence temporarily without any substantial horizontal motion. This coherence loss could be caused by surface flooding, which has been observed to be caused by heavy snow load in this region (Mahoney et al. 2021). Kotzebue Sound also has extensive areas of shallow water ($\sim \leq 2$ m), in which the ice can repeatedly interact with the seafloor as the water level rises and falls under the influence of winds and tides. This process can result in flexural fracturing of the ice surface, which can also lead to coherence loss.

In the month of May, the landfast ice width difference between the two datasets is more consistently negative again, due to extensive areas where no landfast ice was identified by the InSAR-based method, but landfast ice was still present in both the Chukchi and Beaufort coasts in the EM2024 dataset. In particular, Kotzebue Sound never met the coherence threshold to be considered landfast ice during May of any season from 2017–2021. This consistent underestimation by the InSAR-based method is likely due to the alteration of the dielectric properties of the ice surface during the early stages of melt, which leads to the loss of coherence before any substantial motion of the ice.

## 3.2. Monthly mean apparent strain

Although there is considerable variability in monthly mean $\epsilon_a$ at the scale of adjacent pixels, we identify an overall tendency for lower $\epsilon_a$ values to be found near the coast and higher values to occur nearer the SLIE (Fig. 5 and 6). This spatial distribution becomes more evident as the landfast ice season progresses and another tendency emerges whereby the apparent strain in landfast ice tends to decrease over time. By calculating probability distribution of monthly $\epsilon_a$ values, we find that the modal value of $\epsilon_a$ decreases monotonically from December to May (Fig. 7). This indicates that landfast ice becomes more stable the longer it persists.





**Figure 5: Monthly mean apparent strain from 2017–2021 for the months of a) December, b) January, and c) February**







**Figure 6: Monthly mean apparent strain from 2017–2021 for the months of a) March, b) April, and c) May**





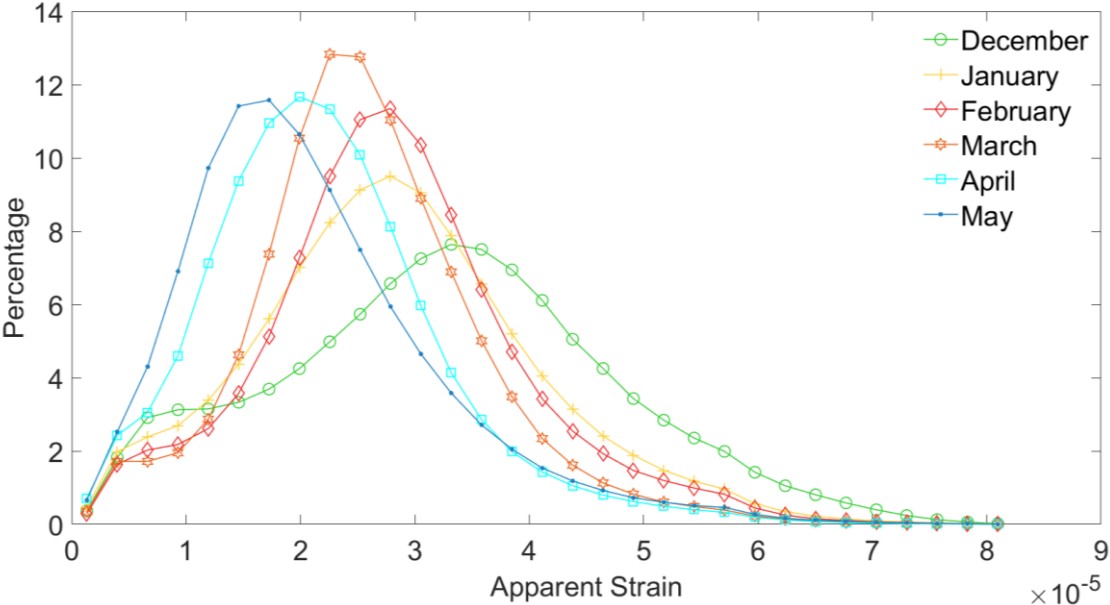

**Figure 7: Distribution of average apparent strain during each month (December, January, February, March, April, and May) between 2017–2021**

### 3.3. Quantitative classification of landfast ice stability

In the Beaufort Sea, landfast ice typically reaches its seasonal maximum width during April (Mahoney et al. 2014). This is also the month for which Dammann et al. (2019) qualitatively defined landfast ice stability throughout the Arctic based on interferometric phase gradient. Hence, we use our mean $\epsilon_a$ values for the month of April to derive probability distributions of $\epsilon_a$ within the three spatial masks identified in Fig. 2, corresponding to bottomfast, sheltered, and not-sheltered ice (Fig. 8). Each distribution has a well-defined and distinct mode, with the modal value for the bottomfast ice mask being the lowest and that for the not-sheltered ice being the highest. The distributions for sheltered and not-sheltered ice are approximately normally distributed, but the bottomfast ice distribution is right skewed, which is the result of high $\epsilon_a$ values being present at the oceanward bottomfast ice boundary, likely associated with tide cracks.

To define the $\epsilon_a$ thresholds that separate the three stability classes initially put forward by Dammann et al (2019) (see section 2.3), we use the lower and upper 10th percentiles of $\epsilon_a$ values found within sheltered ice mask. The lower 10th percentile corresponds to a value of apparent strain of $1.0 \times 10^{-5}$ and closely coincides with the intersection between the sheltered ice and bottomfast ice distributions. $\epsilon_a$ values lower than this are therefore more likely to be found in the bottomfast ice mask than in the sheltered ice mask. The 90th percentile of the sheltered ice distribution corresponds to an apparent strain of $2.3 \times 10^{-5}$. We note that this aligns with the mode of the not-sheltered distribution, rather than intersection between the sheltered and not-



sheltered distributions. However, we expect that areas in the not-sheltered region may still have been stabilized, but by a grounded ridge rather than a permanent barrier island. This likely explains why the mode of the not-sheltered region is broader and is why we did not choose the intersection point of the sheltered and non-sheltered distribution as the threshold between

stabilized and non-stabilized ice. These apparent strain thresholds can now be applied to results from single interferograms to identify areas of varying stability within the landfast ice.

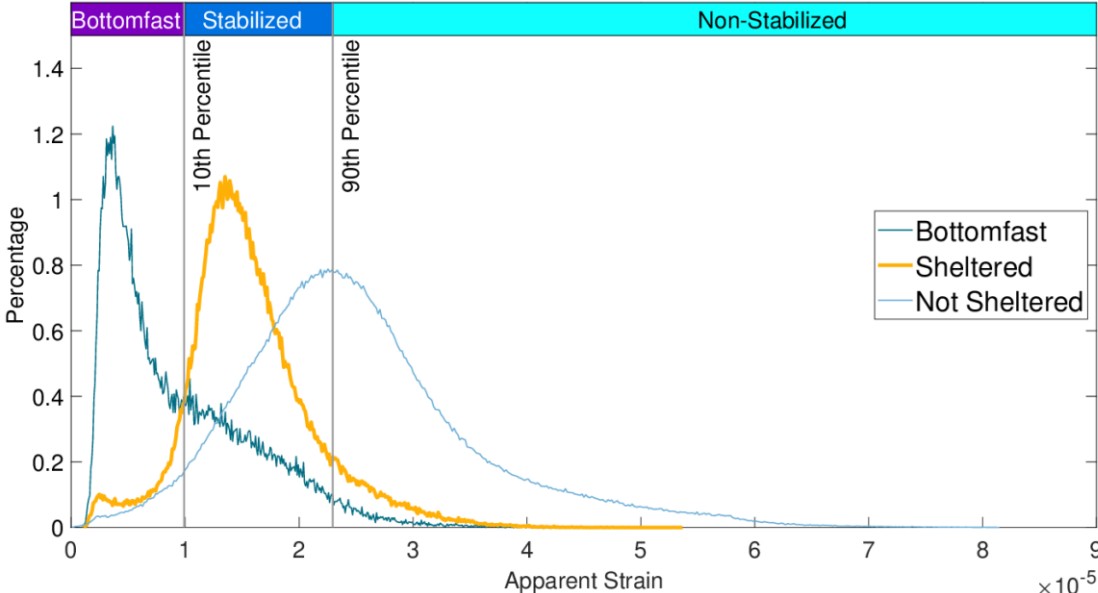

**Figure 8. Distribution of April mean apparent strain, $\epsilon_a$, during the period 2017–2021 in the areas identified as bottomfast, sheltered, and not-sheltered landfast ice.**




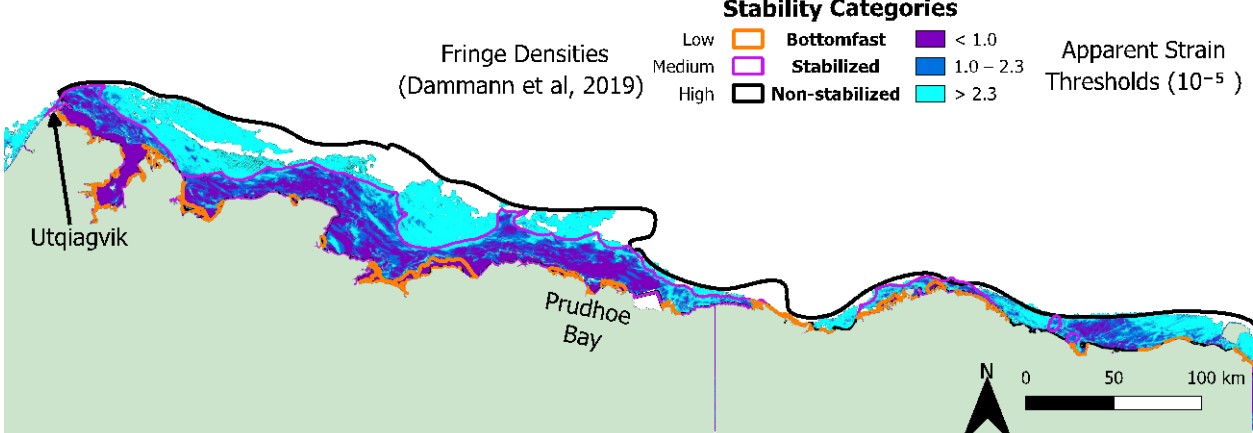

**Figure 9: Categorized landfast ice stability derived from apparent strain threshold applied to interferograms from April 2017 corresponding to those used by Dammann et al. (2019). Orange, purple and black lines indicate the extent of the stability regions defined by Dammann et al (2019), bottomfast, stabilized, and not stabilized. Dark purple, blue and cyan shaded regions correspond to pixels which had apparent strain values within the quantitatively defined stability categories: bottomfast, stabilized and not stabilized.**

We applied the thresholds illustrated in Fig. 8 to $\epsilon_a$ values derived from SAR pairs acquired in April 2017 similar to those used by Dammann et al. (2019) along the Beaufort coast in Alaska. The exact SAR pairs used by Dammann et al. (2019) were not available through ASF's Vertex portal and so we selected overlapping pairs that were acquired within 12 days. This allows us to directly compare the extents of each stability class determined from our $\epsilon_a$ thresholds with those manually delineated by Dammann et al (Fig. 9). Overall, the boundary between stabilized and non-stabilized landfast ice agrees between the methods suggesting that our $\epsilon_a$ threshold of 2.3×10⁻⁵ derived from 4-year monthly averages can be usefully applied to data from individual SAR pairs. Damman et al (2019) did not apply a rigorous coherence threshold and identify a greater extent of landfast ice extent in some areas, but otherwise both methods show good agreement on the position of the SLIE. However, there is a large discrepancy in area identified as bottomfast ice, which we attribute to abnormally low apparent strain in regions behind barrier islands at this particular time. The $\epsilon_a$ values throughout Elson lagoon are considerably lower than the mean value for this month (Fig. 6) and fall within range typically found in bottomfast ice.

## 4. Discussion

### 4.1. Suitability of InSAR for routine identification of landfast sea ice

In general, landfast ice extent is captured well using InSAR during the winter months of the landfast ice season but under-identifies landfast ice extent at the beginning or end of the season. Despite the typical presence of landfast in both the Chukchi and Beaufort regions during November (Mahoney et al. 2024), we did not find any pixels outside the land mask with



a coherence values <0.1 until December, representing a total underestimate of landfast ice during the earliest stage of formation
We attribute the loss of coherence over immobile to the rapid changes in the dielectric properties of the ice surface that occurred
during early stages of growth (e.g., Winebrenner et al. 1996). During the months of December and January, the average the
monthly width of landfast ice identified by our InSAR based method is slightly less than the average monthly landfast ice
width from the EM2024 dataset from 2017–2021 (Fig. 4a-b). Much of the difference in the Beaufort region occurs, between
Point Barrow and Kaktovik where our InSAR-derived results show the landfast ice is consistently 6–10 km narrower. In the
Chukchi region, differences between InSAR-derived width and EM2024 monthly mean width occur primarily in Kotzebue
Sound. Similar to the area between Point Barrow and Kaktovik in the Beaufort region, the InSAR-derived width was on
average 7.7 km less than the EM2024 width in Kotzebue Sound during January.

      Agreement between InSAR-derived landfast ice extent and the EM2024 dataset is best from February through April.
In February and March, our InSAR-based method underestimated landfast ice width compared to the EM2024 dataset by an
average of 2.1 km and 1.1 km, respectively for the combined Chukchi and Beaufort regions. In April the InSAR-derived
landfast ice width exceeded EM2024 width by an average of 0.7 km across both study regions. The main areas where
differences persisted from February through April were Kotzebue Sound and the Colville Delta. Finally, in May, the InSAR
based method under measured the landfast ice width consistently across the study region. The consistent under measuring of
landfast ice in May is attributed to surface melting causing a loss of coherence between acquisitions. Overall, the InSAR based
identification of landfast ice measures the landfast ice width well however the there are certain areas where the 12 days between
acquisitions prevent the methods from identifying landfast ice. On this basis, we find that 12-day repeat Sentinel-1 InSAR may
be a useful tool for helping discriminate landfast ice sea ice extent during the coldest months of the year when the dielectric
properties of the ice surface are most stable. With the imminent launch of the NISAR satellite, 12-day L-band InSAR will be
possible for all but the most northern regions of landfast ice in the Arctic. Meyer et al. (2011) showed that L-band coherence
could be maintained for 45 days over landfast ice and so we anticipate that NISAR may allow us to extend the useful season
of InSAR for this landfast ice detection.

**4.2. Regional variability and annual evolution of landfast ice stability**

      We have shown that, over the study area, the distribution of apparent strain evolves such that the modal value of $\epsilon_a$
decreases monotonically from December to May. We interpret this to indicate that landfast ice becomes more stable the longer
it remains in place over winter. To better understand the processes likely to responsible for this, we partitioned this analysis
between the 11 subregions shown in Fig. 1. In most cases, the mode of the apparent strain distribution decreases monotonically
from month to month, such that landfast ice transitions toward more stable categories over time (Fig. 10). It should be noted
that, from December to April, this increase in stability occurs while the overall extent of landfast ice also increases, suggesting
that the process by which the SLIE advances also increases the stability of the ice overall.

      In some sub regions, the total landfast ice extent in May in less than that in April, so any increase in the modal $\epsilon_a$ value
between these two months likely indicates that the least stable landfast ice is the first to detach. This is especially apparent in



subregion 6 ,Cape Lisburne to Point Hope, where the monthly average $\epsilon_a$ values for May fall almost entirely within the bottomfast and stabilized ice classes (Fig. 10f).

        Despite the overall tendency of $\epsilon_a$ to monotonically decrease over winter, we note that subregions 3, 5, and 7 behaved differently. In subregion 3, Outer Kotzebue Sound, the modal apparent strain decreased until March but remained similar in April. By May, the only landfast ice remaining within Kotzebue Sound was not intersected by any of the coast vectors for this

subregion. Subregion 5, Cape Krusenstern to Point Hope, is the only south facing subregion and exhibited increasing apparent strain from December to March, indicating that landfast ice becomes increasingly unstable in this region as it extends from shore. This suggests the landfast ice along this stretch of coast includes relatively few grounded ridges. However, there is a pronounced shift in the $\epsilon_a$ distribution in April toward increased stability, which we cannot explain with the data available. In subregion 7, Cape Lisburne to Wainwright, the lowest modal value of $\epsilon_a$ was observed in December, when the only landfast

ice present occurred within lagoons. However, the highest modal values of $\epsilon_a$ in this subregion occurred in January and February as landfast ice began forming outside the lagoons. In March, April, and May, $\epsilon_a$ values decreased monotonically in other regions.





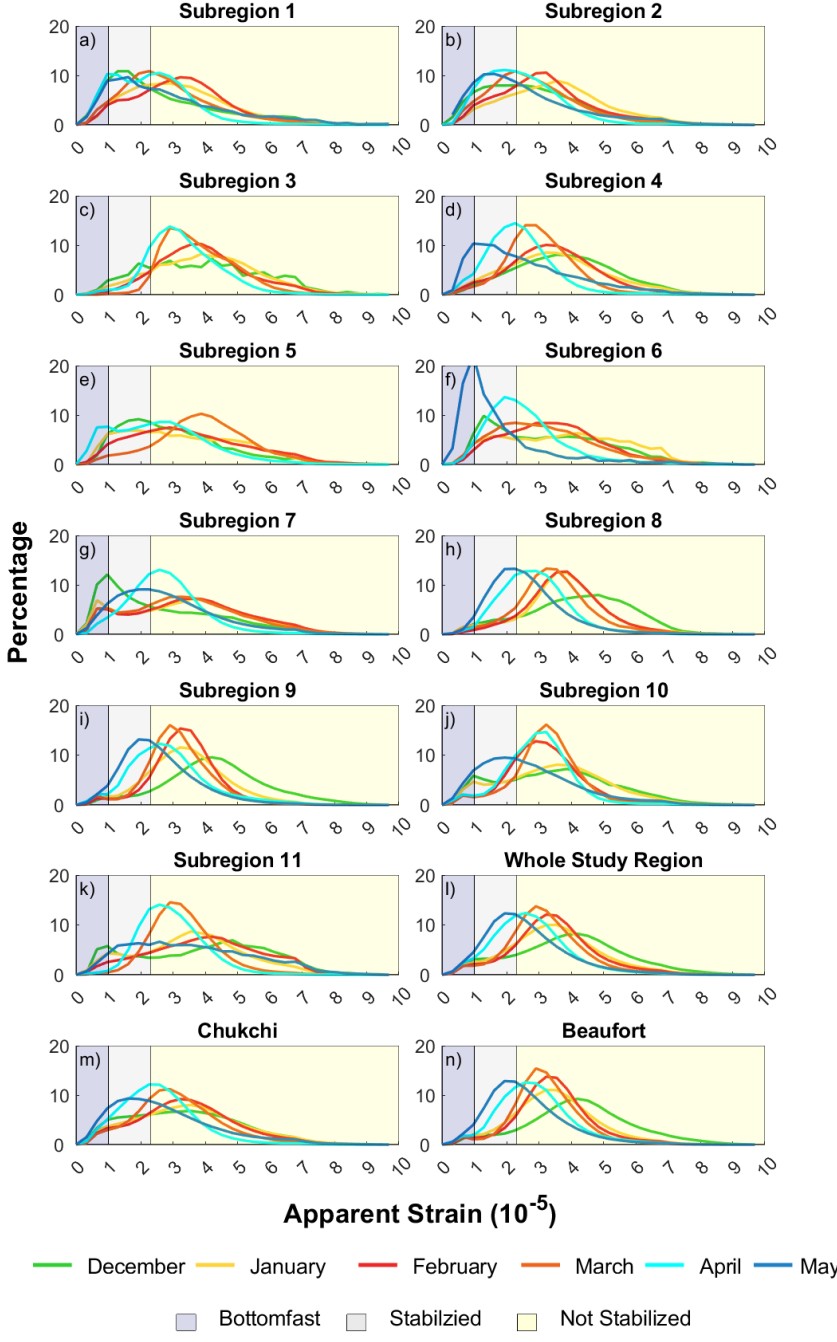

**Figure 10: Monthly distribution of interferometric phase gradient in each of the 2 regions, 11 subregions, and entire study region. Shaded regions indicated the stability classes discussed in section 2.5**





### 4.3. Identification of abrupt increases in apparent strain associated with grounded features

We have shown that the three apparent strain thresholds we defined in section 3.3 for bottomfast, stabilized, and not stabilized landfast ice align well with the extent of equivalent stability regions identified qualitatively by Dammann et al. (2019) from single interferograms. Within other single-interferogram apparent strain maps, we commonly observe strong gradients in apparent strain that are not evident in the monthly average maps illustrated in Figures 5 and 6 which we believe are associated with bottomfast or grounded ice features. For example, at the seaward edge of the bottomfast ice, there is a tide crack zone where the floating landfast ice flexes in response to sea level variations while the bottomfast ice remains stationary. The difference in vertical motion across this narrow region (typically a few 10s of m wide) leads to a region of high fringe density, like those found at the grounding lines of ice shelves and tidewater glaciers (Friedl et al., 2020), resulting in high apparent strain. Examples of such features can be seen in two apparent strain maps from April 05 – 17, 2017 (Fig. 11a) and April 09 – 21, 2022 (Fig. 11b) near the shoreline and around barrier islands in the eastern Beaufort region, denoted by red boxes. In addition, we observe isolated regions of high apparent strain within regions classified as stabilized ice (indicated by cyan boxes in Fig. 11b). The resemblance of these features to those around barrier islands leads us to interpret them as the tide crack zone around grounded ridges. Moreover, they occur at the locations grounded ridges identified in this region during the 2021-22 landfast ice season by Lange et al. (2024).

The shoreward most feature identified in 2022 (Fig. 11b) occurred at approximately the same location as the boundary between the stabilized and not stabilized landfast ice in 2017 (marked by the red line in Fig. 11a) and 2022 seasons. This boundary by another steep gradient in apparent strain, but one which differs from those associated with tide cracks in that it represents a step change between regions of comparatively low and high apparent strain. This is illustrated by looking at apparent strain values along the transect line extending due north from near Oliktok Point, Alaska (Fig. 11c, d). This transect starts at the shoreline and in both years shows elevated apparent strain values as it crosses a barrier island before encountering another peak in apparent strain around 12 km from the shore. In 2017, the apparent strain values seaward of this feature elevated in the range of not stabilized ice (Fig. 11c), but in 2022 they remain lower in the range for stabilized ice until approximately 2 km beyond the seaward most grounded feature. At this point we see a step-change in apparent strain, similar to that at the location of the shoreward ridge in 2017, that marks the boundary between the stabilized and not stabilized ice classes. Hence, although the boundary between stabilized and not stabilized is not always marked by such a sharp gradient in apparent strain, its position in locations like Oliktok Point appears to be controlled in part by the locations of features that we interpret to be grounded ridges.





**Figure 11: Apparent strain values derived from two single-pair interferograms from a) April 05, 2017 – April 17, 2017, and b) April 09, 2022 – April 21, 2022, near Prudhoe Bay, Alaska. The red boxes indicate high apparent strain values either side of a barrier island areas interpreted to be tide cracks along the transect. The cyan boxes panel b) identify similar features, which we interpret as grounded ridges. The apparent strain values along the transects in panels a) and b) are shown in panels c) and d) respectively.**

## 5. Conclusion

InSAR-based methods hold great promise for improving our understanding of both the spatial extent of landfast sea ice and its relative stability. Meyer et al. (2011) already demonstrated the usefulness of interferometric coherence in delineating landfast ice from the mobile pack ice using 45-day repeat L-band PALSAR data. Our investigation shows that the same approach works well during most of the year and throughout most of our study area using 12-day C-band Sentinel-1 data. During the winter months, February – April, landfast ice widths derived from coherence threshold approach typically agreed to within a few km, of those derived from the EM2024 dataset (Fig 4c-e). However, factors unrelated to the motion of the ice



reduce coherence at the beginning and end of the season, occasionally resulting in false absence of landfast ice (corresponding to a 100% underestimate in Fig 4e and f, for example). Hence, the usefulness of out interferometric coherence method for identifying landfast ice would be improved with either a shorter period between SAR images or a use of longer radar wavelength. We therefore look forward to the launch of the NASA-ISRO synthetic aperture radar (NISAR) currently scheduled

to launch in 2025. The L-band sensor and 12-day repeat interval should open new opportunities for routine mapping of landfast extent using InSAR.

In a quantitative expansion of Dammann et al's (2019) work, we find that apparent strain, $\epsilon_a$, can serve as a meaningful measure landfast ice stability. Specifically, we find that the distributions of $\epsilon_a$ values within the bottomfast ice zone and either seaward or shoreward of barrier islands have distinct modes. This allows us to define three landfast ice stability classes on the

basis of apparent strain: bottomfast ice ($\epsilon_a < 1.0 \times 10^{-5}$), stabilized ($1.0 \times 10^{-5} \leq \epsilon_a < 2.3 \times 10^{-5}$), and not stabilized ($\epsilon_a \geq 2.3 \times 10^{-5}$). Single interferograms indicate a spatially abrupt change in apparent strain between the stabilized and not stabilized landfast ice. These abrupt changes are not present within the monthly average apparent strain images (Figs. 5 and 6) which suggests they do not commonly persist between interferograms. This is likely because the strain field in landfast ice evolves more rapidly than the repeat interval of Sentinel-1 and because the monthly averages combine results from different

viewing geometries. Nonetheless, the boundary between stabilized and not stabilized ice is often associated with features that we interpret to be grounded features (Fig. 11), which suggests our classification represents physical difference in the strain experienced by landfast ice on either side. To our knowledge, this is the first time InSAR has been used to identify individual grounded ridges based on localized values of high apparent strain or phase gradient. Our interpretation on this matter is strongly supported by matching the locations of grounded ridges identified by Lange et al. (2024), based on ICESat-2 altimetry data.

Moreover, our observations support their findings that grounded ridges are commonly located shoreward of the landfast ice edge

Along with the establishment of the apparent strain thresholds we defined, we observed a decrease in apparent strain throughout the season. The increase in stability the longer the landfast ice is likely due to thickening, but further investigation into the cause is needed. A fundamental flaw in using information derived from interferometry is the inability of InSAR to

capture along-track deformation. Being limited to the magnitude of line-of-sight deformation causes an underrepresentation of the total possible deformation which occurred. Implementing methods to resolve the two-dimensional strain, demonstrated by Fedders et al. (2024), would improve our ability to classify landfast ice stability based on the apparent strain. The ability of InSAR to identify these areas of varying stability can hold immense value to members of Arctic coastal communities such that they can continue to operate on the landfast ice safely.



**6. Code Availability**

Code for processing of the ASF Vertex outputs into apparent strain can be found here (https://github.com/aheinhorn/Alaska_InSAR_CODE). All the code for analyzing the fast ice extent of the InSAR derived width and EM2024 derived width can be found here (https://github.com/armahoney/SLIEalyzer)

**7. Data Availability**

Sentinel-1 SAR imagery are openly accessible at the Alaska Satellite Facility's Vertex tool (https://search.asf.alaska.edu), where users can search for imagery, request on-demand processing (include InSAR products), and download data. Registration with NASA EarthData is required but is open to anyone. The EM2024 dataset is available through the University of Alaska Fairbanks Scenarios Network for Alaska and Arctic Planning (SNAP) at http://data.snap.uaf.edu/data/Base/Other/Landfast_Sea_Ice/Chukchi_Daily (for the Chukchi region) and

http://data.snap.uaf.edu/data/Base/Other/Landfast_Sea_Ice/Beaufort_Daily (for the Beaufort region) under a Creative Commons license (CC-BY 4.0)

**8. Author Contribution**

All authors edited the manuscript. AHE led the analysis, produced all figured except Fig. 1, and drafted the paper. ARM produced Fig. 1 and provided scientific and editorial feedback and direction to the project.

**9. Competing Interests**

The authors declare that they have no conflict of interest.

**10. Disclaimer**

The views and conclusions contained on the website are those of the authors and should not be interpreted as representing the opinions or policies of the US Government, nor does mention of trade names or commercial products constitute endorsement

or recommendation for use.

**11. Acknowledgments**

Study collaboration and funding were provided by the US Department of the Interior, Bureau of Ocean Energy Management (BOEM), Environmental Studies Program, Washington, DC, under Agreement Number M19AC00021. We are also extremely grateful to the members of our Science Review Board, Hajo Eicken, Andrew Roberts, and John Walsh



## 12. Financial Support

Study collaboration and funding were provided by the US Department of the Interior, Bureau of Ocean Energy Management (BOEM), Environmental Studies Program, Washington, DC, under Agreement Number M19AC00021




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
