# Peer review of "Wintertime Evolution of Landfast Ice Stability in Alaska from InSAR"

_EGUsphere, 2025_

## Referee Comment (RC2)

*Wintertime Evolution of Landfast Ice Stability in Alaska from InSAR*
**Author(s): Andrew Harrison Einhorn and Andrew Richard Mahoney**
**Egusphere-2025-567**

The article presents a novel quantitative method for identifying stable regions of landfast sea ice and provides an empirical differentiation of bottom fast, stable, and unstable fast ice using Interferometric Synthetic Aperture Radar (InSAR). Landfast sea ice is the immobile ice that is anchored to the land. In the Arctic during winter, this ice connects coastal communities. The authors utilise coherence and the phase gradient of various InSAR pairs over the Chukchi and Beaufort Seas in the Arctic. Their research encompasses data gathered from 2017 to 2021, specifically during the winter months when the sea ice forms.

The article highlights two main points: First, it delineates the width of landfast sea ice areas based on the coherence of the InSAR pairs. While this concept is not entirely new, the novelty lies in the comparison with a new dataset to benchmark the results. Second, the authors employ the phase gradient of the interferometric signal to assess the stability of landfast sea ice, focusing on the apparent strain in the satellite's line of sight.

The most significant finding of the study is the differentiation of bottom fast, stable, and unstable landfast sea ice through InSAR apparent strain, along with the mapping of its evolution over time and space. Overall, this article advances the interpretation and analysis of landfast sea ice using radar remote sensing techniques.

This article is interesting, well-written, and clear. It enhances our understanding of the stability of landfast sea ice and offers an alternative perspective on its progression throughout the season. It would be intriguing to explore whether this methodology could be applied in other regions and if varying viewing angles could provide additional insights into sea ice deformation.

**Comments regarding the overall text:**

Overall, this is an interesting article that deserves to be published. However, a few issues need to be addressed before it is ready for publication.

All figures, except for Figure 11, have been utilised in Mahoney et al. (2024), which is a report to the US Bureau of Ocean Energy Management. Since this report is not a peer-reviewed paper, it presents a missed opportunity to provide a more in-depth explanation of the EM2024 methodology. This could be accomplished by either enhancing the text in section 2.2 or including it as supplementary material. The EM2024 dataset and its methodology could greatly benefit the scientific community studying Arctic regions.

The article presents solid results that are compared to other datasets. However, comparing these datasets quantitatively and statistically, when possible, would be beneficial. For instance, how do the EM2024 and InSAR-based datasets differ in terms of distance or area? What are the percentage differences, and how do these differences evolve throughout the seasons? Are these two datasets statistically comparable? While they likely are, providing these numbers would strengthen the case for using one or the other to define the Landfast sea-ice area in the future. The same applies to the areas of bottom fast, stable, and unstable

fast ice when comparing apparent strain rates to those reported by Dammann et al. (2019) in Figure 9. Although comparisons are made and conclusions drawn, statistical values are lacking. Presenting these numbers would reinforce the argument for using relative strains rather than solely relying on the InSAR fringe approach. For example, in section 4.3, the first statement claims that the apparent strain thresholds align well with those of Dammann et al. (2019). Please include a statistical comparison to support this. Additionally, it would be interesting to explore the quantitative behaviour of stabilised versus sheltered and non-sheltered areas; a simple table could help visualise this information.

Finally, comparing the apparent strain data with a bathymetry map of the region would be valuable.

**Comments on the methods:**
The data section states that 14 reference scenes were used to derive the data. Was there an effort to utilize different SAR geometries (ascending and descending paths) over the same area to compare the relative strain results? What criteria were used to select those 14 IW reference scenes? Were they chosen solely for their coverage, or was there prior knowledge about the area involved?

For instance, could the different regional behaviors observed in Figure 10 be partially attributed to the geometry of the scenes? Would it be more effective to identify the optimal line of sight (LOS) for assessing the stability of landfast sea ice?

**Comments on figures:**
In general, the figures are well done and have all the necessary components. However, for someone unfamiliar with the area, it takes some time to orient themselves between the figures and the corresponding names.

Figure 1 -  Since this is the first figure and shows the study area, it would be helpful to add a rectangle that outlines the area related to Figure 2. This can be done with a dashed line to avoid interfering with the other information displayed in the figure. The caption should clearly state that the different shades of blue represent various contiguous areas as defined in the figure. Additionally, instead of "coast vectors," I recommend using "coast normal vectors."

Figure 2 -  Is "Utqiagvik" the same as "Point Barrow"? Utqiagvik is not mentioned in the text, so it would be better to remain consistent with "Point Barrow." This figure illustrates the masks used to define the different landfast sea ice areas, which should be stated in the caption. It is important to geo-reference this area with the larger study region. One way to do this is to add a rectangle in Figure 1 that overlaps the same area, and the other option is to include an inset that showcases the larger region alongside the example area.

Figure 3 - Please add reference names such as Chukchi Sea, Beaufort Sea, Point Barrow, Prudhoe Bay, Kotzebue Sound, and/or others.

Figure 4 - "Kotzebue Sound" appears twice on the x-axis, which seems like a typo. If intentional, it should be defined differently (e.g., "Kotzebue 2") and indicated on the map in Figure 3 or 1.

Figure 5 and 6 - Add a star or another marker to show where the transect in Figure 11 is located. In the caption for Figure 11, only the area near Prudhoe Bay is mentioned. Please clarify what the dashed oval in Figure 6b represents. Is that "Elson Lagoon," as referred to in line 630?

Figures 7 and 8 - Are the y-axes in Figures 7 and 8 the same in terms of percentages? It is unclear whether they represent distributions of the same total. Would it be possible to plot the total distribution of landfast sea ice for April (as shown in Figure 7) in a light color or with 50% transparency in the background of Figure 8? This would help illustrate the 10th and 90th percentiles, that comes from April overall data, correct?

Figure 9 - It is unclear whether the apparent strain values used in this figure are derived from April 2017 data or represent an average of April data from all years, as seen in Figure 8. Please clarify in the caption.

Figure 11 - The transect used in this figure could be added to Figure 9 to provide a spatial understanding of its location.

**In-text comments:**
Overall, the article is well written, and I have only a few minor comments:

Throughout the text, "Landfast sea-ice" is referred to either as "Landfast sea-ice" or "Landfast Ice." I recommend using "Landfast sea-ice" consistently or, after the first introduction, switching to "fast-ice," with a phrasing such as: "hereafter referred to as fast-ice.

Line 89 - Please provide examples of processes that could reduce coherence, such as snowfall, surface reworking by wind, or melting.

Line 102 - The assumption that variation of the phase gradient in LOS dominate the surface motion is key for the developed of the apparent strain methodology. However, I am not completely sure if this can be said so bluntly, a more extensive explanation is needed here. Please rephrase this sentence, talk about the importance of the relative surface motion of landfast sea-ice and explain how this relative measurement can be a pertinent way of defining stability in Landfast sea-ice.

Line 130 - Please rewrite, the last part of the sentence, "...bottomfast ice will be found in waters up to approximately 1.5 m deep", to something like "..bottomfast ice is normally found…"

Line 131 - Please add the citation (Dammann et al., 2019) to the sentence that defines bottomfast sea-ice mapping from InSAR fringes.

Line 155 - Please check that the sentence is correct, "from these" looks out of place.

Line 239 - The word "the" is repeated after "average".

Line 242 - Add Point Barrow and Katkovik to Figure 3 and reference here. This applies to other significant areas mentioned in this section as well.

Line 268 - Change ".. in May in.." for "...in May is…"

---

## Author Comment (AC1)

**Response to Reviewers' Comments**
*Manuscript egusphere-2025-567*

**Wintertime Evolution of Landfast Ice Stability in Alaska from InSAR**

by Andrew H. Einhorn and Andrew R. Mahoney

**Overview**

Below, we provide our responses explaining how we have addressed each of the reviewers' comments. For clarity and ease of reference, we have broken some comments into individual points. Our responses to all comments are highlighted in blue. Descriptive comments that we feel did not include specific points that required addressing are italicized in grey text.

**Reviewer #1 comments**

*The authors present a study that proposes an InSAR coherence threshold to be used with Sentinel-1 image pairs to find the extent of landfast sea ice on the north coast of Alaska. They mention a loss of coherence in at the beginning and end of the winter season. Most of the study is an extension of Damman et al. (2019), with the current study finding 'apparent strain' values that separates landfast sea ice into three categories: bottom-fast ice, stabilized ice, and not-stabilized ice. The authors describe nuances related to the location of the category thresholds and the seasonal evolution of strain values. The efficacy of the methods is evaluated against a climatology for landfast sea ice extent.*

**Major points**

The Introduction is rather brief. Please include material defining and relating stability, strain, and displacement, in the context of sea ice. Please also include a survey of similar InSAR techniques for sea ice, including the references listed below, and particularly a more-detailed description of the works on which this study is built (i.e., Dammann et al. 2019; Meyer et al. 2011; Pratt 2022).

- Dammert, P. B. G., Lepparanta, M., & Askne, J. (1998). SAR interferometry over Baltic Sea ice. International Journal of Remote Sensing, 19(16), 3019-3037.
- Li, S., Shapiro, L., McNutt, L., & Feffers, A. (1996). Application of satellite radar interferometry to the detection of sea ice deformation. Journal of the Remote Sensing Society of Japan, 16(2), 153-163.
- Morris, K., Li, S., & Jeffries, M. (1999). Meso-and microscale sea-ice motion in the East Siberian Sea as determined from ERS-1 SAR data. Journal of Glaciology, 45(150), 370-383.
- Wang, Z., Liu, J., Wang, J., Wang, L., Luo, M., Wang, Z., ... & Li, H. (2020). Resolving and analyzing landfast ice deformation by InSAR technology combined with Sentinel-1A ascending and descending orbits data. Sensors, 20(22), 6561.

Thank you for pointing out we need to include more content relating to sea ice and more specifically landfast ice stability. In addition to adding content relating to the 3 articles mentioned, we plan on adding more content in the introduction relating to the 3 papers which formed the foundation of this work: Meyer et al. (2011), Dammann et al. (2019), and Pratt

(2022). These more thorough descriptions of the mentioned journal articles, in addition to other supplementary texts, will provide the readers with a complete understanding of landfast ice stability, the techniques previously used to observed stability, and the utilization of InSAR in the sea ice field.

The map figures in Figures 1, 2, 3, 5, 6 and 9 are inconsistent in the application of standard map elements. Use a consistent lat/lon grid, scalebar type, land colour, and shadow zone colour throughout.
We will create a standard style used in all maps and figures. The style will use similar colors to represent similar meaning (Ex. stability categories having the same color on a map and plot figures). The style will be CVD-friendly.

The use of the 0.1 coherence threshold value found by Meyer et al. (2011) for L-band is not well-supported for C-band. Coherence values for C-band are usually significantly higher for sea ice. The assumption of an adequate trade-off between temporal baselines and wavelengths may be true, but this should be supported with evidence. A sensitivity study related to the Figure 4 results may help. Please also provide a representative example of a coherence image alongside a calibrated SAR image, showing landfast sea ice and open water or mobile sea ice. The coherence values should be presented in such a way that they are easy to discern.
You are correct that we did not substantiate our choice of the 0.1 coherence threshold. Our plan is to take a subset of our data, 5-10 images from the Chukchi region and 5-10 from the Beaufort region from various months and apply different coherence thresholds to identify landfast ice. We can then compare the position of the seaward landfast ice edge identified as landfast ice by each threshold. In addition, when we chose the 0.1 threshold, we did some general testing to confirm our choice and noticed when increasing the threshold above 0.1 the areas identified as landfast ice constrained a considerable number of holes. These localized areas of low to zero coherence are still considered landfast according to our definition. We also noticed the landfast ice edge did not change considerably when adjusting the threshold. Regardless we need to substance these observations within this article. We will also include the representative example of a coherence image alongside the calibrated SAR image as requested.

The use of the 90$^{th}$ percentile (Figure 8) as the threshold between stabilized and not-stabilized is not convincing. It seems to be based on a rather vague notion that some of the not-stabilized is actually stabilized. This is presented without any supporting evidence. The 10$^{th}$ percentile threshold is also not well-supported. A more robust method for finding the thresholds should be applied. There are many statistical methods to choose from. Given the mix of distribution types, a non-parametric technique such as a decision tree is recommended.
We appreciate the constructive criticisms of our methodology. We believe that this comment is the result of poor explanation of the methodology. We believe the miscommunication occurred when differentiating between sheltered vs stabilized landfast ice. We will better define and differentiate sheltered landfast ice from stabilized in the updates of the article. However, we would like to provide some text provide context and a better explanation. When deriving the apparent strain thresholds we created 3 classes: bottomfast, sheltered, and not sheltered. These classes were not based on any observations we made. Instead, the extents of each class were either defined by a pervious study, Dammann et al. (2019) for bottomfast ice, or geomorphologic characteristics of the coastline. The extent of the sheltered landfast ice polygon (orange polygons in figure 2) indicates areas of the coastline which have barrier islands oceanward of

that position. The positions of these barrier islands are known and negligible between seasons. Therefore, we know that whatever landfast ice forms within the bounds of the orange polygons will have a grounded feature, the barrier island, oceanward of its position. Barrier islands perform 2 roles for the landfast ice formed shoreward of the barrier island, sheltering the fast ice from dynamic forces such as wave and currents and provides an offshore anchor point or stabilizing feature. Similarly, grounded ridges provide an offshore stabilizing feature for fast ice shoreward of its position. The ridges will mitigate some of the dynamic forces, such as attenuation of wave energy, but does not shelter the fast ice the same way the barrier islands do. Conceptually sheltered fast ice fast ice that is shoreward of a barrier island where the fast ice is sheltered from dynamic forces and stabilized by the barrier island where stabilized fast ice is just stabilized by an offshore grounded ridge.

Our theory when coming up with a way to determine the apparent strain thresholds for the stability categories used the following thought process. We know that grounded ridges do not always form in the same location each season thus we cannot identify a line of grounded ridges that would accurately delineate stabilized fast ice from not-stabilized inter-seasonally. We also know that barrier islands provide a similar role to grounded ridges in terms of stabilizing and the location of the barrier islands does not change inter-seasonally. When looking at Figure 2, any fast ice which forms in the sheltered region, orange polygons, is guaranteed to have a stabilizing feature offshore of its position in the form of a barrier island. Within the blue cross filled regions fast ice formed here it is possible the landfast ice formed here has an offshore stabilizing feature, in the form of a grounded ridge, but it is also possible there is not a grounded ridge offshore of its position. It is this nuance that separates sheltered from stabilized and why we define the upper threshold of the stabilized fast ice as the 90th percentile of the sheltered fast ice apparent strain. One last important distinction is that the classes bottomfast, sheltered, and not sheltered are defined by the location of where the fast ice resided where the stability categories bottomfast, stabilized, and non-stabilized are based on apparent strain values.

We will incorporate better and more thorough explanations of these nuances and within the updated text. We will include explicit definitions of the stability categories, and the classes used to derive the stability thresholds.

However, it seems that the use of monthly averages is obscuring the abrupt change from stabilized to not stabilized strain values, as can be observed in the individual scenes in Figure 11. Perhaps a more meaningful method can be found to estimate the strain thresholds, based on individual images and the coast vectors.

This is a great suggestion. We plan to explore a methodology which utilizes single apparent strain maps to derive the thresholds. However, we will still use apparent strain maps from the month of April as the fast ice has the best coherence and the most fast ice using this method along the Beaufort coastline.

Overall, the paper needs to substantiate the coherence and strain thresholds further. It may be better to localize the analysis to two or three sub-regions, and investigate these in more detail, as was done in Figure 11, in concert with air temperature data. This may lead to a better understanding of what is affecting the coherence and strain values, and lead to better estimates for the strain thresholds.

We see the value in having localized analysis. We do not believe the thresholds will vary greatly if the analysis local. With regards to incorporating air temperature into our analysis, which we

believe the reviewer is meaning strain associated with thermal contraction/expansion of the fast ice we are doubtful adding air temperature to our analysis will improve our understanding. An analysis that includes air temperature and uses air temperature in concert with interferometric phase changes in a 2D inverse modeling method was done by Fedders et al (2024). Their methodology excluded interferograms with curved fringe patterns and their study area was limited to Elson lagoon, a sheltered embayment where the fast ice is not exposed to significant dynamic forces. In Elson Lagoon the deformation of fast ice is dominated by thermal forces. Since our analysis includes areas which are dominated by both dynamic and thermal forces. Without the ability to assume what the dominant forces are and that they vary across the study area it is not possible to use the 2D method demonstrated by Fedders et al (2024). In line with a related minor comment, we plan to create a confusion matrix which compares how pixels were categorized by Dammann et al (2019) and our stability categories to validate our thresholds.

The analysis of the strain images to identify grounded ridges is a useful element of this study, and one that could be expanded upon by analyzing more image pairs of smaller regions. We agree with the need for more data, however the data included in the analysis is all the data made available to the authors. We included over 2000 Sentinel-1 A/A or B/B pairs in our analysis and utilized all data available to us.

Minor points:
Line 52: In Figure 1, the eleven smaller areas should be briefly mentioned in the text, or if they are unimportant, they can be removed from Figure 1. Indicate what the difference is between the cyan and blue vectors.
The authors realize the ambiguity of the numbers and colors contained in Figure 1. We will improve the figure and supporting text to clearly indicate that the numbers in figure 1 represent the subregions in Figure 10. The colors there are shaded regions the align with the regions described in Figures 4 and 10. We will improve the coloring of the coast vectors as we mistakenly used the same colors to differenced adjacent subregions as the shaded regions denoting the regions.

Line 66: If the coast-normal vectors are conceptual in Figure 1, please indicate that in the caption. Otherwise, indicate the time period that the vectors represent.
The coast vectors represented in Figure 1 are every $10^{th}$ coast vector. We will clarify this in the caption and apply corrections associated with the colours from previous comments regarding Figure 1.

Line 88: Please describe the "other processes unrelated to motion that reduce coherence", with references.
We will add text describing the processes which are not associate with sea ice drift which can affect the coherence. Specifically, we will discuss surface melt, snow drift, and surface deformation.

Line 93: The grammar needs to be improved for this sentence.
We will adjust the grammar to the following: Following the work of Meyer et al. (2011), we apply the same coherence threshold to identify landfast ice using C-band Sentinel-1 imagery,

under the assumption that the increased coherence resulting from a shorter repeat orbit interval at least partially offsets the reduced coherence associated with the shorter wavelength.

Line 146: How can the extent of the 'not sheltered ice' be known a priori, in order to create a mask for it. Also, it is not clear what these masks will be used for.
In line with previous major comments, we will improve our description and distinction of the sheltered and not sheltered regions. To directly address this comment, we defined the sheltered fast ice region as any area where landfast ice forms which has a barrier island offshore of its location. By deduction then any area where fast ice forms which a barrier island does not have offshore of its position is considered Not Sheltered. The offshore edge of the Not Sheltered was created such that all fast ice identified using the coherence threshold is accounted for.

We will also improve the clarity of text associated with the masks represented in Figure 2 to explain how they were defined and how they are used.

Line 160: The Figure 3 caption references Figure 1 regarding the 'shadow' zones. However, these zones are not indicated or obvious in Figure 1. Also, what do the shadow zones represent? Please indicate the Chuckchi-Beaufort border in Figure 3.
The reviewer is correct. We mistakenly omitted the shadow zones within Figure 1. We will add these regions to Figure 1. These shadow zones are areas where our coast vectors do not reach, mainly associated with complex coastline shapes. The coast vectors cannot cross a landmass which is not an island. For example, Admiralty Bay, ~50 km east of Barrow is a shadow zone as the coast vectors originate at the shoreline of Elson Lagoon. Additionally, Near Kotzebue the coastline is quite complex and since the coast vectors cannot cross any non-island landmass areas such as Hotham Inlet and Selawik Lake are classed as shadow zone. We will also add similar region boundaries from Figure 1 to Figure 3.

Line 163: In Figure 4, the x-axis text is too large, with the location names bleeding into one another. Also, these locations do not seem to align with the sub-regions in Figure 1. There are two Kotzebue locations on the x-axis.
We will alter the x-axis labels such that they do not overlap. The labels represent villages or coastline features, not the sub-regions. We will alter these labels to be oriented to the centre coast vector within each subregion and have the subregion name.

Line 172: In Figure 4e, the high variability in extent just east of the Chuckchi-Beaufort border is not evident in Figure 3, which is also for April. Please explain the inconsistency between figures.
The variability at the boundary between the Chukchi and Beaufort regions is associated with Kotzebue Sound. The order of the coast vectors is a bit jumbled but will be sorted properly in the edits.

Line 191: Please explain how the percentage values (y-axis in Figures 7, 8, and 10) are calculated. What does percentage represent, especially in Figure 8? The percentages do not appear to add up when comparing Figures 7 and 8. Also indicate in the Figure 7 and 8 captions, the region the data represent.
Thank you for pointing out the confusion. Figure 7 is the distribution of both the Chukchi and Beaufort regions for each month while Figure 8 is just from April in the Beaufort region. The bins are spaced differently in Figures 7 and 8 to provide a clearer result in Figure 7. The larger bin sizes smooth the distribution and allows for easier distinction between the months. Figure 7

becomes cluttered when a similar bin window to Figure 8 is used. We found it important to depict the noise in Figure 8.

Line 201: In Figure 8, please add an overall distribution so that the reader can see if the modes are present in the overall un-masked data.
We can add the total distribution to Figure 8. This will not be the same distribution as plotted in Figure 7 (cyan with squares) as it will just be the Beaufort region. We do see the value in having these data plotted on Figure 8 and will add to the plot.

Line 224: Is this single April 2017 comparison the only validation for the proposed thresholds for stability classes? If so, then the evidence is not convincing enough to say that the proposed threshold "…can be usefully applied…". The large areas seemingly misclassified as bottom fast ice are adjacent to much of the not-stabilized ice. This juxtaposition does not support the statement that "the boundary between stabilized and non-stabilized landfast ice agrees between methods". Furthermore, the outer extent of the not-stabilized ice is significantly greater in Dammann et al. (2019), which does not support the statement "both methods show good agreement on the position of the SLIE". This comparative analysis should be redone with additional data. Given the delineations from Dammann et al. (2019) it is reasonable to include a quantitative comparison, e.g., a confusion matrix.
The review is correct that this the only validation for the thresholds. We will produce a confusion matrix as suggested to compare how our classifications aligned with Dammann et al. (2019). With the different method for the threshold derivation using single SAR pairs instead of the monthly average we are confident the thresholds will produce a confusion matrix where the majority of pixels are classified the same using both methods.

Line 230: Is the 'particular time' four years of April data? If so, then are the data for this region's stabilized ice included in the distributions in Figure 8? If this is an anomalous area, then why not use a more representative area?
"this particular time" refers to the SAR pairs acquired from April of 2017 used by Dammann et al (2019) and this study. We will rephrase to clarify this statement.

Line 237: Should this not be > 0.1?
Yes. Thank you for point this out. We will correct the text to mean greater than 0.1.

Line 240: Provide a quantity instead of 'slightly less'.
We believe the best statistic to quantify the difference would be a percent difference of the fast ice width at each coast vector. We will calculate this value and likely include mean value for each month in a table.

Line 250: Where is the Colville Delta?
We will update all maps to include better labels of places and features references within the text.

Line 252: Please provide the May air temperatures to corroborate the attribution.
We can provide general idea of the air temperate in May and how this likely lead to surface melting. In addition, we will show the degradation of the coherence from April to May and June do depict why we believe the cause is surface melting not deformation.

Line 253: It is not clear how or where the 12-day repeat prevents landfast ice identification.

Thank you for pointing this out. This is an awkwardly phrased sentence that we will improve. We were trying to indicate that with a 12-day orbit interval the combination of C-Band and the surface melt cause the loss of coherence. The way we posed the sentence this way was there is shorter orbit intervals, and we believe this would allow for better coherence values during the spring season.

Line 261: Refer to Figure 7 in the first sentence.
We will refer for Figure 7 in the first sentence of this paragraph.

Line 265: In Figure 10, should not Figure 10l (whole study region) be the same as Figure 7? The values and monthly peaks are different. Also, is 'interferometric phase gradient' in the caption supposed to mean 'apparent strain'?
Figure 10l and Figure 7 contain the same data. The y-axis limit is different which could cause a perceived difference between the plots.
The reviewer is correct that in the caption "interferometric phase gradient" should be Apparent Strain.

Line 269: Is the extent not a function of coherence, which is indicated to be poor in May. Does this affect the pdf results shown for May?
This brings about an interesting point about defining landfast ice and the method and criteria needed to be classes as landfast ice. You will note that in Figure 4 the coherence-based method for identifying fast ice consistently underrepresented the amount of fast ice compared to the EM2024 dataset in May. We observed that in June the coherence-based identification of fast ice did not work due to suspected surface melting. The method struggles to accurately identify fast ice once melting occurs at the surface. This is a bias with this method and why we limited our analysis to the winter months (December-May).
While we identified the coherence during the month of May as poor, there were still areas which met our coherence-based requirements. We believe that the poor coherence during May is acceptable since we are aware of the bias. We will include addition text to ensure that we fully describe that shortcoming of the method. May differs from June as there is still extensive areas identified as fast ice during May.

Line 287: The statement that the apparent strain threshold 'work well' has yet to be shown. This may need to be revised in light of previous comments.
After the new methods and threshold evaluation are conducted, we will reassess this sentence. We will also ensure that we do present how well the thresholds work (i.e. confusion matrix).

Line 301: In Figure 11's caption, indicate that the thick red line is the boundary between the stabilized and not stabilized landfast ice. Indicate the location of the red line in panels c and d as well.
We will implement the boundary between Stabilized and Not Stabilized from panels a and b into panel c and d.

Line 304: Some of the text in Figure 11c and d is too small.
We will alter Figure 11 to ensure all text is an appropriate size and legible.

Line 302: The sentence needs to be reworded: "This boundary by another steep…".

Thank you for pointing this out. The sentence should read "This boundary is marked by another steep gradient in apparent strain, but one which difference from those associated with tide crack in that it represents a step change between regions of comparatively low and high apparent strain."

Line 320: The factors unrelated to motion causing a loss of coherence could be investigated in this study.
In response to other comments, we plan to investigate air temperature and the associated thermal expansion and contraction of the landfast ice.
Line 323: Why would a shorter period improve coherence if the cause is temperature and snow moisture related? An analysis of the air temperature would likely provide some evidence towards a cause.
The shorter orbit time simply provides less time between acquisitions for the surface to change (melt or snow event). With a shorter period between acquisitions there is less opportunity for these processes to degrade the signal.

Line 343: The thickening of the ice is a reasonable assumption as to the cause of the seasonal decrease in apparent strain. This should be investigated in this study, using air temperature data and a simple freezing model.
We can implement a simple freezing degree day model to approximate the ice thickness over the season from 2017-2021 to justify our hypothesis that the reduction in apparent strain throughout a season is associated with thickening of the landfast ice.

Line 347: Why was a 2-D method not pursued?
The 2D model demonstrated by Fedders et al. (2024) had differences compared to this study which made a similar analysis not possible. The study area for Fedders et al. (2024) was Elson Lagoon. Based on the coastal morphology or Elson Lagoon and the requirement that the fringe patterns had to be simple, Fedders et al. (2024) was able to assume that the dynamic forces such as waves, winds, and pack ice interactions were small and the dominate forces deforming the fast ice were thermal forces. Since our study contained fast ice where we did not know if dynamic or thermal forces were dominating the 2D method is not possible.

**Reviewer 2 Comments to the Author**

**General**
*The article presents a novel quantitative method for identifying stable regions of landfast sea ice and provides an empirical differentiation of bottom fast, stable, and unstable fast ice using Interferometric Synthetic Aperture Radar (InSAR). Landfast sea ice is the immobile ice that is anchored to the land. In the Arctic during winter, this ice connects coastal communities. The authors utilise coherence and the phase gradient of various InSAR pairs over the Chukchi and Beaufort Seas in the Arctic. Their research encompasses data gathered from 2017 to 2021, specifically during the winter months when the sea ice forms.*

*The article highlights two main points: First, it delineates the width of landfast sea ice areas based on the coherence of the InSAR pairs. While this concept is not entirely new, the novelty lies in the comparison with a new dataset to benchmark the results. Second, the authors employ the phase gradient of the interferometric signal to assess the stability of landfast sea ice, focusing on the apparent strain in the satellite's line of sight.*

*The most significant finding of the study is the differentiation of bottom fast, stable, and unstable landfast sea ice through InSAR apparent strain, along with the mapping of its evolution over time and space. Overall, this article advances the interpretation and analysis of landfast sea ice using radar remote sensing techniques.*

*This article is interesting, well-written, and clear. It enhances our understanding of the stability of landfast sea ice and offers an alternative perspective on its progression throughout the season. It would be intriguing to explore whether this methodology could be applied in other regions and if varying viewing angles could provide additional insights into sea ice deformation.*

**Comments regarding the overall text**

*Overall, this is an interesting article that deserves to be published. However, a few issues need to be addressed before it is ready for publication.*

All figures, except for Figure 11, have been utilised in Mahoney et al. (2024), which is a report to the US Bureau of Ocean Energy Management. Since this report is not a peer-reviewed paper, it presents a missed opportunity to provide a more in-depth explanation of the EM2024 methodology. This could be accomplished by either enhancing the text in section 2.2 or including it as supplementary material. The EM2024 dataset and its methodology could greatly benefit the scientific community studying Arctic regions.

The EM2024 dataset, creation and analysis, is more thoroughly described in a "The Evolving Decline of Landfast Sea Ice in Northern Alaska and Adjacent Waters: from an Updated Climatology" by the same authors of this study. At the time of submission, the paper detailing the EM2024 dataset had not been submitted and was not citable. Since then, the paper has been submitted and is under review.

The article presents solid results that are compared to other datasets. However, comparing these datasets quantitatively and statistically, when possible, would be beneficial. For instance, how do the EM2024 and InSAR-based datasets differ in terms of distance or area? What are the percentage differences, and how do these differences evolve throughout the seasons? Are these two datasets statistically comparable? While they likely are, providing these numbers would strengthen the case for using one or the other to define the Landfast sea-ice area in the future. The same applies to the areas of bottom fast, stable, and unstable fast ice when comparing apparent strain rates to those reported by Dammann et al. (2019) in Figure 9. Although comparisons are made and conclusions drawn, statistical values are lacking. Presenting these numbers would reinforce the argument for using relative strains rather than solely relying on the InSAR fringe approach. For example, in section 4.3, the first statement claims that the apparent strain thresholds align well with those of Dammann et al. (2019). Please include a statistical comparison to support this. Additionally, it would be interesting to explore the quantitative behaviour of stabilised versus sheltered and non-sheltered areas; a simple table could help visualise this information.

In response to this comment, we can add a more qualitative analysis of the difference between the EM2024 dataset and the InSAR-derived fast ice. Instead of using area we will opt to use the fast ice extent, defined as the distance along coast-normal vectors.

The idea to do a similar analysis for the bottomfast, stabilize, and not stabilized fast ice classes is a great idea. We will attempt to implement some qualitative analysis describing the evolution of the amount of each stability class of landfast ice exists from each SAR pair.

We plan to create a confusion matrix which will compare the classes defined by Dammann et al. (2019) and our stability classes on a pixel-by-pixel basis.

Finally, comparing the apparent strain data with a bathymetry map of the region would be valuable.

We can include some analysis analyzing the apparent strain values in each 10 m isobath. We would expect fast ice in waters deeper than 20 m will be classified as not stabilized as grounded ridges typically do not form deeper than 20 m (Mahoney et al. 2007).

**Comments on the methods:**

The data section states that 14 reference scenes were used to derive the data. Was there an effort to utilize different SAR geometries (ascending and descending paths) over the same area to compare the relative strain results? What criteria were used to select those 14 IW reference scenes? Were they chosen solely for their coverage, or was there prior knowledge about the area involved.

The 14 reference scenes were chosen purely for their coverage of the study area. Through the Alaska Satellite Facility Vertex Portal, we designated criteria including the temporal baseline, IW mode, and spatial baseline and were presented with references scenes where we could choose pairs of SAR acquisitions to process using the HyP3 algorithm. We will add a figure which depicts the reference scene coverage in the supplementary material.

For instance, could the different regional behaviors observed in Figure 10 be partially attributed to the geometry of the scenes? Would it be more effective to identify the optimal line of sight (LOS) for assessing the stability of landfast sea ice?

We acknowledge that the acquisition geometry can influence the direction of motion that InSAR is sensitive too. In our study, we did not explicitly control ascending vs descending orbits. However, we believe that we cannot assume all surface motion occurs in the same direction with respect to the coastline. This assumption is what led to the creation of the term apparent strain where we recognize the strain, we are measuring is the minimum amount of strain that has occurred over the 12-day period. InSAR is not sensitive to along-track deformation which for both ascending and descending orbits is approximately north-south.

If we were to assume all surface motion happens in a perpendicular direction to the coastline, compression or expansion, then we would not be able to resolve the deformation for the subregions with predominantly north or south aspect. We assume the deformation is occurring in all directions including vertical.

**Comments on figures:**

In general, the figures are well done and have all the necessary components. However, for someone unfamiliar with the area, it takes some time to orient themselves between the figures and the corresponding names.

We will make efforts to better orient readers to the study area. We will add inset maps to certain figures and labels of villages and some geographic features.

Figure 1 - Since this is the first figure and shows the study area, it would be helpful to add a rectangle that outlines the area related to Figure 2. This can be done with a dashed line to avoid interfering with the other information displayed in the figure. The caption should clearly state that the different shades of blue represent various contiguous areas as defined in the figure. Additionally, instead of "coast vectors," I recommend using "coast normal vectors."

We will improve this figure in multiple ways. Instead of having the coast normal vectors colors in alternating shaded of blue and cyan to differentiate each subregion from the adjacent ones we will adopt a color scheme that has a unique color associated with each subregion. We also plan to adopt the term coast normal vectors for this figure and in text.

We also plan to use this figure as a general orientation for other figures by including dashed lines indicating the boundaries of other figures and how they fit into the study region.

Figure 2 - Is "Utqiagvik" the same as "Point Barrow"? Utqiagvik is not mentioned in the text, so it would be better to remain consistent with "Point Barrow." This figure illustrates the masks used to define the different landfast sea ice areas, which should be stated in the caption. It is important to geo-reference this area with the larger study region. One way to do this is to add a rectangle in Figure 1 that overlaps the same area, and the other option is to include an inset that showcases the larger region alongside the example area.

Utqiagvik is the traditional name of the village formally known as Barrow. We will include text to differentiate Utqiagvik (Barrow) and Point Barrow. We will orient readers within the study region by including a dashed line within Figure 1. We will also update the caption to describe what these regions are used for and ensure that the text describes how and why these regions were defined.

Figure 3 - Please add reference names such as Chukchi Sea, Beaufort Sea, Point Barrow, Prudhoe Bay, Kotzebue Sound, and/or others.

We will add geographical markers such as villages or named coastal features to orient the readers.

Figure 4 - "Kotzebue Sound" appears twice on the x-axis, which seems like a typo. If intentional, it should be defined differently (e.g., "Kotzebue 2") and indicated on the map in Figure 3 or 1.

Figure 4 is going to be reorganized, and the x-axis labels will be associated with the subregion names instead of villages.

Figure 5 and 6 - Add a star or another marker to show where the transect in Figure 11 is located. In the caption for Figure 11, only the area near Prudhoe Bay is mentioned. Please clarify what the dashed oval in Figure 6b represents. Is that "Elson Lagoon," as referred to in line 630?

We will add geographic markers to these figures to orient the reader. We will opt to add an inset map to Figure 11 to indicate where in the study region the transect is located.

Figures 7 and 8 - Are the y-axes in Figures 7 and 8 the same in terms of percentages? It is unclear whether they represent distributions of the same total. Would it be possible to plot the total distribution of landfast sea ice for April (as shown in Figure 7) in a light color or with 50% transparency in the background of Figure 8? This would help illustrate the 10th and

90th percentiles, that comes from April overall data, correct?

Figures 7 and 8 have different bin sizes where Figure 8 has 10 times the number of bins. We found that using the same number of bins as Figure 8 in Figure 7 cluttereing the figure. We also found that including grouping the distribution in Figure 8 into the smaller bins has value.

We are planning on adding the total distribution for the Beaufort region during April to Figure 8.

Figure 9 - It is unclear whether the apparent strain values used in this figure are derived from April 2017 data or represent an average of April data from all years, as seen in Figure 8. Please clarify in the caption.

Thank you for pointing out the need for improved clarity with Figure 9. Figure 9 is meant to provide a visual comparison of the spatial extent of the stability regions defined by Dammann et al. (2019) vs the quantitative defined regions on the same SAR pairs. To provide improved clarity we will add the dates of the SAR pair acquisitions and the spatial extent for each of these SAR pairs. The SAR pairs are not all from the same dates, thus we generalized the SAR pairs as having occurred during April of 2017.

Figure 11 - The transect used in this figure could be added to Figure 9 to provide a spatial understanding of its location.

To orient the readers within Figure 11 we are going to add inset maps of Alaska with boxes indicating where the transect is located. We believe adding the transect location onto Figures 5,6, or 9 would be misleading as the transect does not use the apparent strain maps depicted in those figures.

**In-text comments:**
*Overall, the article is well written, and I have only a few minor comments:*

Throughout the text, "Landfast sea-ice" is referred to either as "Landfast sea-ice" or "Landfast Ice." I recommend using "Landfast sea-ice" consistently or, after the first introduction, switching to "fast-ice," with a phrasing such as: "hereafter referred to as fast-ice.

We will use the consistent term of either fast ice or fast-ice when refereeing to landfast ice within the text.

Line 89 - Please provide examples of processes that could reduce coherence, such as snowfall, surface reworking by wind, or melting.

We will include other examples of processes which reduce coherence not associated with sea ice drift.

Line 102 - The assumption that variation of the phase gradient in LOS dominate the surface motion is key for the developed of the apparent strain methodology. However, I am not completely sure if this can be said so bluntly, a more extensive explanation is needed here. Please rephrase this sentence, talk about the importance of the relative surface motion of landfast sea-ice and explain how this relative measurement can be a pertinent way of defining stability in Landfast sea-ice.

We apricate the reviewer's suggestion to clarify a key assumption of our methodology. We will rephrase the original sentence and add additional text to enhance clarity. The Line-of Sight-phase is directly proportional to the amount of line-of-sight strain which has occurred between the satellite acquisitions. For a medium such as landfast ice the relative motion (i.e. the differential displacement between neighboring areas) is indicative of mechanical instability and strength. For example, if the surface of one area of fast ice is moving differently than an adjacent this will result in internal strain and produce cracks potentially causing detachment of the fast ice. The build up of internal strain and creation of cracks resulting in detachment are all signs of instability and unsafe fast ice.

Line 130 - Please rewrite, the last part of the sentence, "...bottomfast ice will be found in waters up to approximately 1.5 m deep", to something like "..bottomfast ice is normally found…"
We will rewrite the sentence to "This depends on the ice thickness (or, more precisely, the draft of the ice), but in our study region bottomfast ice is normally found in waters shallower than 1.5 m (Pratt 2022)."

Line 131 - Please add the citation (Dammann et al., 2019) to the sentence that defines bottomfast sea-ice mapping from InSAR fringes.
We will add the citation to this sentence

Line 155 - Please check that the sentence is correct, "from these" looks out of place.
Thank you for pointing this out. This is a mistake made by the authors. "from these" will be removed.

Line 239 - The word "the" is repeated after "average".
This is a typo. Thank you for making the authors aware of this mistake.

Line 242 - Add Point Barrow and Katkovik to Figure 3 and reference here. This applies to other significant areas mentioned in this section as well.
We will add annotation to Figure 3 which highlights the area mentioned in addition to geographic markers within Figure 3.

Line 268 - Change ".. in May in.." for "...in May is…"
We appreciate your diligence and catching our grammatical mistakes.